# *Mir155* regulates osteogenesis and bone mass phenotype via targeting *S1pr1* gene

Zhichao Zheng[1,2†], Lihong Wu[1†], Zhicong Li[1†], Ruoshu Tang[1], Hongtao Li[3], Yinyin Huang[1], Tianqi Wang[1], Shaofen Xu[1], Haoyu Cheng[1], Zhitong Ye[1], Dong Xiao[4,5], Xiaolin Lin[4,5], Gang Wu[6,7*], Richard T Jaspers[1,2*], Janak L Pathak[1*]

[1]Affiliated Stomatology Hospital of Guangzhou Medical University, Guangdong Engineering Research Center of Oral Restoration and Reconstruction, Guangzhou Key Laboratory of Basic and Applied Research of Oral Regenerative Medicine, Guangzhou, China; [2]Laboratory for Myology, Department of Human Movement Sciences, Faculty of Behavioural and Movement Sciences, Vrije Universiteit Amsterdam, Amsterdam Movement Sciences, Amsterdam, Netherlands; [3]State Key Laboratory of Respiratory Diseases, National Clinical Research Center for Respiratory Diseases, Guangzhou Institute of Respiratory Health, the First Affiliated Hospital of Guangzhou Medical University, Guangzhou, China; [4]Guangdong Provincial Key Laboratory of Cancer Immunotherapy Research and Guangzhou Key Laboratory of Tumour Immunology Research, Cancer Research Institute, School of Basic Medical Science, Southern Medical University, Guangzhou, China; [5]Institute of Comparative Medicine & Laboratory Animal Center, Southern Medical University, Guangzhou, China; [6]Department of Oral and Maxillofacial Surgery/Pathology, Amsterdam UMC and Academic Center for Dentistry Amsterdam (ACTA), Amsterdam Movement Science, Vrije Universiteit Amsterdam, Amsterdam, Netherlands; [7]Department of Oral Cell Biology, Academic Center for Dentistry Amsterdam (ACTA), University of Amsterdam and Vrije Universiteit Amsterdam, Amsterdam, Netherlands

**\*For correspondence:**
g.wu@acta.nl (GW);
r.t.jaspers@vu.nl (RTJ);
j.pathak@gzhmu.edu.cn (JLP)

†These authors contributed equally to this work

**Competing interest:** The authors declare that no competing interests exist.

**Abstract** MicroRNA-155 (miR155) is overexpressed in various inflammatory diseases and cancer, in which bone resorption and osteolysis are frequently observed. However, the role of miR155 on osteogenesis and bone mass phenotype is still unknown. Here, we report a low bone mass phenotype in the long bone of *Mir155*-Tg mice compared with wild-type mice. In contrast, *Mir155*-KO mice showed a high bone mass phenotype and protective effect against inflammation-induced bone loss. *Mir155*-KO mice showed robust bone regeneration in the ectopic and orthotopic model, but *Mir155*-Tg mice showed compromised bone regeneration compared with the wild-type mice. Similarly, the osteogenic differentiation potential of bone marrow stromal stem cells (BMSCs) from *Mir155*-KO mice was robust and *Mir155*-Tg was compromised compared with that of wild-type mice. Moreover, *Mir155* knockdown in BMSCs from wild-type mice showed higher osteogenic differentiation potential, supporting the results from *Mir155*-KO mice. TargetScan analysis predicted sphingosine 1-phosphate receptor-1 (*S1pr1*) as a target gene of *Mir155*, which was further confirmed by luciferase assay and *Mir155* knockdown. *S1pr1* overexpression in BMSCs robustly promoted osteogenic differentiation without affecting cell viability and proliferation. Furthermore, osteoclastogenic differentiation of *Mir155*-Tg bone marrow-derived macrophages was inhibited compared with that of wild-type mice. Thus, *Mir155* showed a catabolic effect on osteogenesis and bone mass phenotype via interaction with the *S1pr1* gene, suggesting inhibition of *Mir155* as a potential strategy for bone regeneration and bone defect healing.

## Editor's evaluation

The authors have shown the important role of miR155 in promoting bone regeneration and higher bone mass. Their fundamental work shows compelling evidence that miR155 is overexpressed in inflammatory diseases therefore, the use of anti-miR155 could produce anti-inflammatory effects. This shows the importance of miR155 inhibitors as therapeutics to promote bone regeneration in inflammatory conditions.

## Introduction

MicroRNAs (miRNAs) are a class of endogenous non-coding RNAs with 18–22 nucleotides length that bind to the 3'-untranslated region of the target gene and regulate the target gene expression (*Gareev et al., 2020*). miRNAs regulate cell functions such as growth, differentiation, and energy metabolism by silencing the target gene via degradation or translational repression (*Gareev et al., 2020*; *Wang et al., 2019b*). Moreover, miRNAs are also involved in the pathophysiology of various inflammatory diseases and cancers (*Kumar et al., 2017*; *Acunzo et al., 2015*; *Gao et al., 2020*). Certain miRNAs had been reported to regulate osteogenesis and bone homeostasis (*Gao et al., 2020*; *Wang et al., 2019a*). MicroRNA-155 (miR155) is one of the best conserved and multifunctional miRNAs that regulate several biological processes and diseases such as tumorigenesis, cardiovascular disease, kidney diseases, etc. (*Elton et al., 2013*; *Readhead et al., 2020*; *Bala et al., 2016*; *Wu et al., 2018*). miR155 is upregulated in inflammatory diseases and cancers, including periodontitis, lung cancer, liver cancer, and breast cancer (*Wu et al., 2021*; *Shao et al., 2019*; *Xin et al., 2020*; *Pasculli et al., 2020*). Systemic bone loss is frequently observed in patients with inflammatory diseases and cancers (*Dimitroulas et al., 2013*; *Badri et al., 2019*). However, the role of miR155 on osteogenesis and bone homeostasis is still unclear.

Induced osteoclasts formation/activity and compromised osteogenic differentiation disrupt bone homeostasis causing bone loss (*Kitaura et al., 2020*; *Li et al., 2014*). Osteoclast formation and activity are induced during inflammation and cancer (*Adamopoulos, 2018*; *Roodman, 2001*). *Mir155* had been reported to induce osteoclastogenesis (*Kagiya and Nakamura, 2013*). *Mir155* knockout (*Mir155*-KO) mice exhibit reduced local bone destruction in arthritis attributed to reduced generation of osteoclasts (*Blüml et al., 2011*). Osteogenic differentiation of precursor cells results in bone formation and is the key anabolic event of bone homeostasis. Reduced osteogenic differentiation of precursor cells causes low bone mass phenotype increasing the risk of fracture. Osteogenesis is also a key biological process of bone tissue engineering. Various miRNAs targeted approaches have been developed to promote bone regeneration and bone defect healing during bone tissue engineering (*Arriaga et al., 2019*). The role of *Mir155* in osteogenic differentiation and bone regeneration has been rarely investigated. Compromised osteogenesis and low bone mass phenotype are frequently observed in patients with inflammatory diseases and cancers (*Schmidt et al., 2019*; *Mann et al., 2009*). Similarly, effective bone regeneration and bone defect healing are also key challenges in patients with inflammatory diseases. *Mir155* targets multiple genes to regulate the pathophysiology of a specific disease in a cell type-specific manner (*Hsin et al., 2018*). Sphingosine 1-phosphate receptor-1 (*S1pr1*) is one of the target genes of *Mir155* (*Xin et al., 2015*), which has been reported to positively regulate the osteogenic differentiation of precursor cells (*Sato et al., 2012*; *Higashi et al., 2016*). Therefore, it is wise to explore the involvement of *S1pr1* in the *Mir155*-mediated effect on osteogenesis.

In this study, we aimed to analyze the effect of different levels of *Mir155* on osteogenesis and bone mass phenotype using *Mir155* transgenic (*Mir155*-Tg) and *Mir155*-KO mice. This study also investigated the role of *Mir155* target gene *S1pr1* on the osteogenic differentiation of bone marrow stromal stem cells (BMSCs). We found a catabolic effect of *Mir155* on osteogenesis and bone mass phenotype by targeting the *S1pr1* gene.

## Results

### *Mir155*-Tg mice showed a low bone mass phenotype

We analyzed the expression pattern of *Mir155*-Tg mice. *Mir155* expression was 8.57-fold higher in bone tissue of *Mir155*-Tg mice compared with the wild-type mice (*Figure 1A*). Hematoxylin and

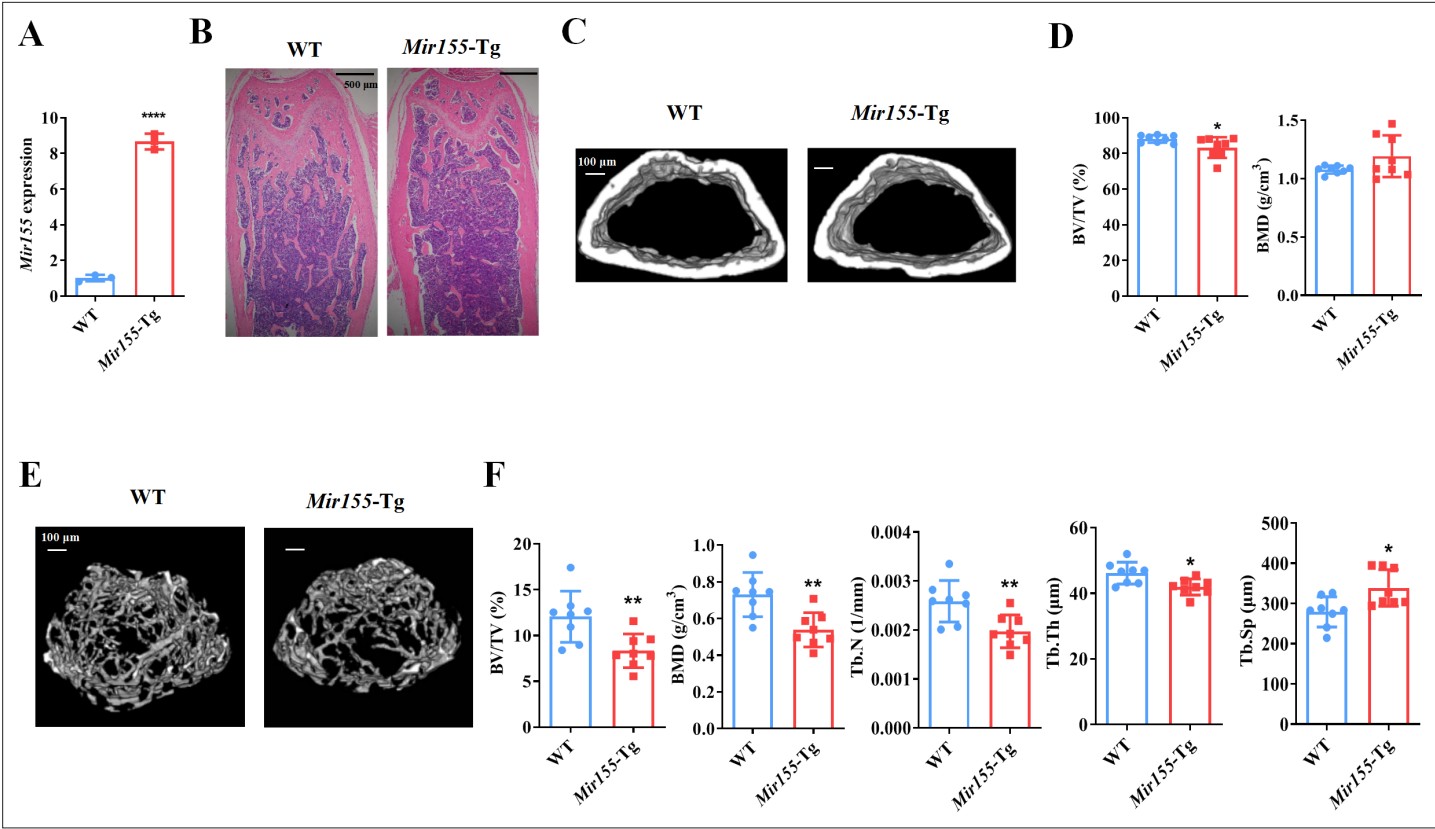

**Figure 1.** *Mir155* transgenic (*Mir155*-Tg) mice showed a low bone mass phenotype. (**A**) *Mir155* expression in bone, (**B**) *Hematoxylin and eosin (H&E) staining*, (**C**) representative micro-CT images for cortical bone, (**D**) bone volume/total volume (BV/TV) and bone mineral density (BMD) analysis, (**E**) representative micro-CT images for trabecular bone, (**F**) BV/TV, BMD, trabecular number (Tb.N), trabecular thickness (Tb.Th), and trabecular separation (Tb.Sp) analysis. Data are presented as mean ± SD, n=8. Significant difference compared to wild-type mice, *p<0.05, **p<0.01, and ****p<0.0001.

The online version of this article includes the following source data for figure 1:

**Source data 1.** Raw data for *Figure 1A, D* (BV/TV and BMD), and *Figure 1F* (BV/TV, BMD, Tb.N, Tb.Th, and Tb.Sp).

eosin (H&E) staining shows the growth plate, trabecular bone, cortical bone, and marrow structure of *Mir155*-Tg and wild-type mice. Low trabecular density was observed in *Mir155*-Tg mice compared with wild-type mice (*Figure 1B*). Micro-CT results showed reduced cortical bone thickness and bone volume/total volume (BV/TV) in *Mir155-Tg* mice (*Figure 1C and D*). The bone mineral density (BMD) level was not significantly changed (*Figure 1D*). Micro-CT images showed fewer and thinner trabeculae in *Mir155*-Tg mice femur compared with wild-type mice (*Figure 1E*). Trabecular bone parameter BV/TV, BMD, trabecular number (Tb.N), and trabecular thickness (Tb.Th) were significantly reduced in *Mir155*-Tg mice compared with wild-type mice (*Figure 1F*). In *Mir155*-Tg mice, trabecular separation (Tb.Sp) was significantly increased compared with wild-type mice (*Figure 1F*). These results indicate the low bone mass phenotype in *Mir155*-Tg mice.

### *Mir155*-KO mice showed a high bone mass phenotype

*Mir155* expression in *Mir155*-KO mice was dramatically downregulated compared with wild-type mice (*Figure 2A*). H&E staining demonstrated that trabeculae were increased in the *Mir155*-KO mice compared with wild-type mice (*Figure 2B*). Micro-CT results showed a higher cortical bone thickness and BV/TV in *Mir155*-KO mice (*Figure 2C and D*). The BMD level was not significantly increased in *Mir155*-KO mice compared with wild-type (*Figure 2D*). Micro-CT images showed robustly dense and interconnected trabeculae in *Mir155*-KO mice compared with wild-type mice (*Figure 2E*). The trabecular bone parameters BV/TV, BMD, and Tb.N in *Mir155*-KO mice were increased by 2-, 2.69-, and 1.83-fold respectively, compared with wild-type mice (*Figure 2F*). While

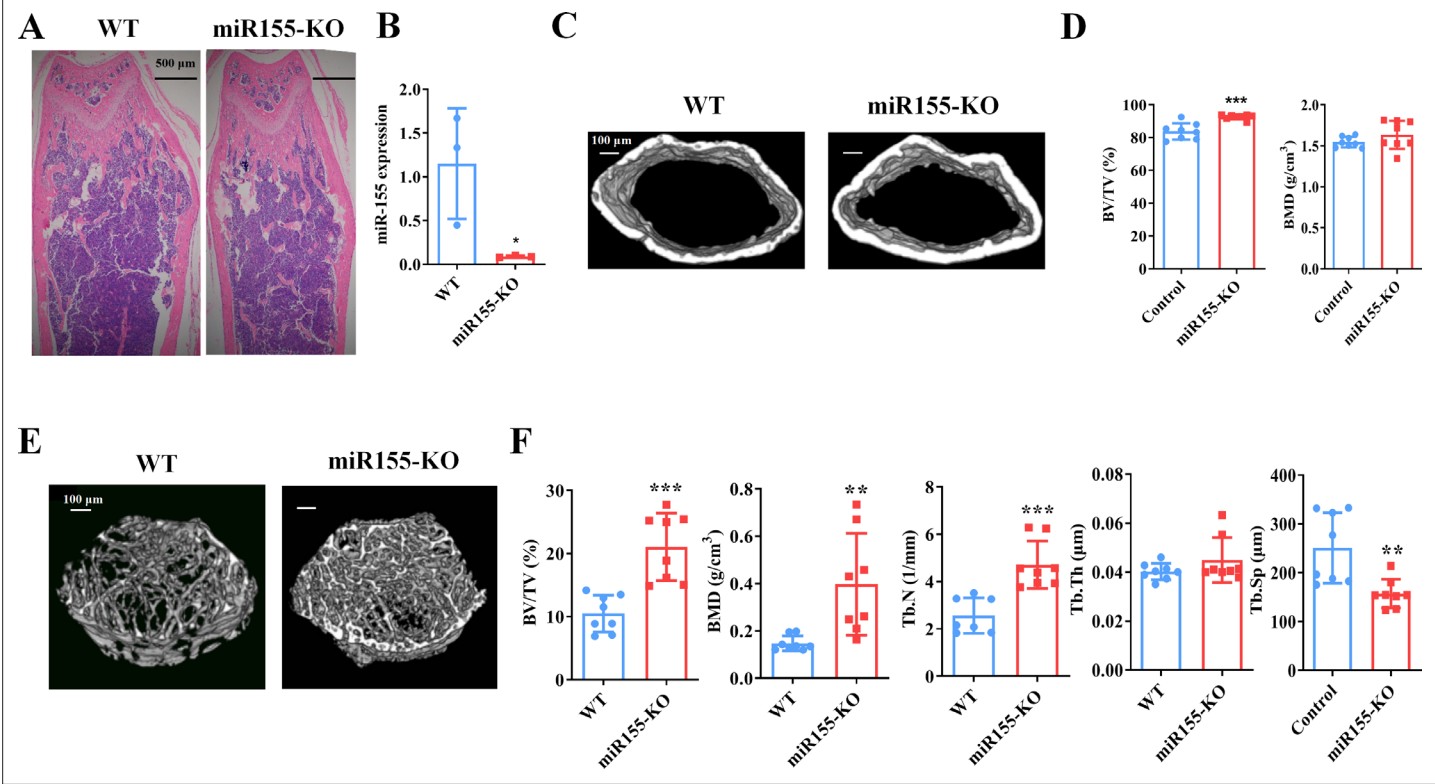

**Figure 2.** *Mir155* knockout (*Mir155*-KO) mice showed a high bone mass phenotype. (**A**) *Mir155* expression in bone, (**B**) Hematoxylin and eosin (H&E) staining, (**C**) representative micro-CT images for cortical bone, (**D**) bone volume/total volume (BV/TV) and bone mineral density (BMD) analysis, (**E**) representative micro-CT images for trabecular bone, (**F**) BV/TV, BMD, trabecular number (Tb.N), trabecular thickness (Tb.Th), and trabecular separation (Tb.Sp) analysis. Data are presented as mean ± SD, n=8. Significant difference compared to wild-type group, *p<0.05, **p<0.01, and ***p<0.001.

The online version of this article includes the following source data for figure 2:

**Source data 1.** Raw data for *Figure 2A, D* (BV/TV and BMD), and *Figure 2F* (BV/TV, BMD, Tb.N, Tb.Th, and Tb.Sp).

Tb.Th was similar in *Mir155*-KO and wild-type mice (*Figure 2F*). Tb.Sp in *Mir155*-KO mice was significantly reduced compared with wild-type mice (*Figure 2F*). These results indicate the high bone mass phenotype of *Mir155*-KO mice. *Mir155*-KO and *Mir155*-Tg showed an opposite trend of bone mass phenotype and bone parameters (*Figures 1 and 2*), suggesting the role of *Mir155* in bone homeostasis regulation.

## Lipopolysaccharide-related osteolysis

We analyzed the effect of *Mir155*-KO in the context of inflammation-related bone loss. H&E staining of long bone tissue sections showed more trabecular bone in lipopolysaccharide (LPS)-treated *Mir155*-KO mice compared with LPS-treated wild-type mice (*Figure 3A*). Micro-CT images showed slightly thicker cortical bone in LPS-treated *Mir155*-KO mice compared with LPS-treated wild-type mice (*Figure 3B*). Cortical bone parameters BV/TV and BMD were significantly increased in LPS-treated *Mir155*-KO mice compared with LPS-treated wild-type mice (*Figure 3C*). Micro-CT images showed more dense and interconnected trabeculae in LPS-treated *Mir155*-KO mice compared with LPS-treated wild-type mice (*Figure 3D*). The trabecular bone parameters BV/TV, BMD, and Tb.N were increased by 2.3-, 1.22-, and 2.68-fold, respectively, in LPS-treated *Mir155*-KO mice compared with LPS-treated wild-type mice (*Figure 3E*). Tb.Sp was significantly decreased in LPS-treated *Mir155*-KO mice compared with LPS-treated wild-type mice (*Figure 3E*). These results demonstrate the protective effect of *Mir155*-KO against LPS-induced bone loss.

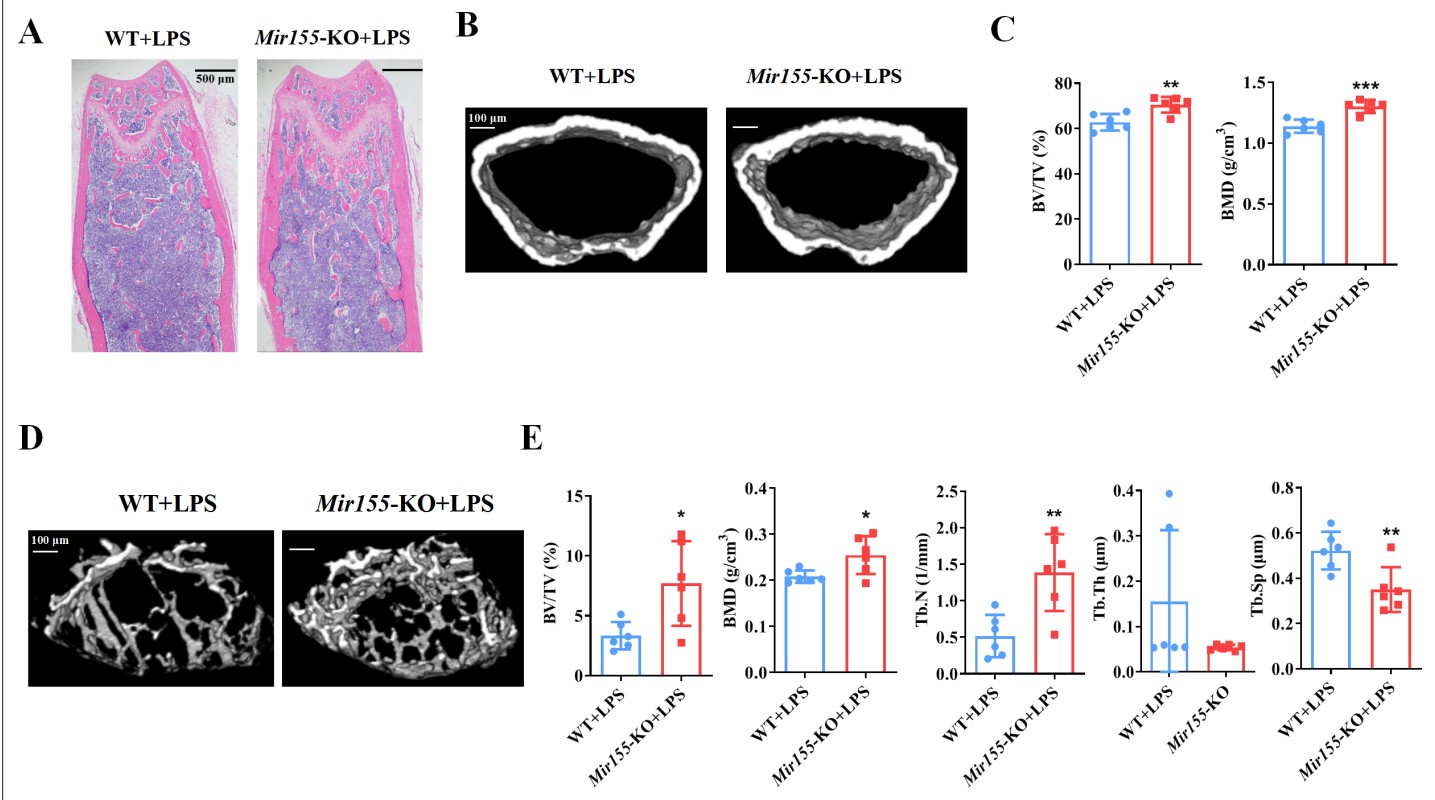

**Figure 3.** *Mir155* knockout (*Mir155*-KO) mice showed higher resistance against lipopolysaccharide (LPS)-induced bone loss. (**A**) Hematoxylin and eosin (H&E) staining, (**B**) representative micro-CT images for cortical bone, (**C**) bone volume/total volume (BV/TV) and bone mineral density (BMD) analysis, (**D**) Representative micro-CT images for trabecular bone, (**E**) BV/TV, BMD, trabecular number (Tb.N), trabecular thickness (Tb.Th), and trabecular separation (Tb.Sp) analysis. Data are presented as mean ± SD, n=6. Significant difference compared to wild-type group, *p<0.05, **p<0.01, and ***p<0.001.

The online version of this article includes the following source data for figure 3:

**Source data 1.** Raw data for *Figure 3C* (BV/TV and BMD) and *Figure 3E* (BV/TV, BMD, Tb.N, Tb.Th, and Tb.Sp).

### Ectopic bone regeneration was inhibited in *Mir155*-Tg mice while increased in *Mir155*-KO mice

A bone regeneration study was conducted to investigate whether the bone regeneration potential is altered in *Mir155*-Tg and *Mir155*-KO mice. BMP2-loaded collagen membranes were implanted in mice ectopically to confirm the bone regeneration potential in the ectopic site of *Mir155*-Tg, *Mir155*-KO, and the respective wild-type mice. Micro-CT images showed very less bone volume in collagen membrane transplanted in *Mir155*-Tg mice compared with that of wild-type mice (*Figure 4A*). The *Mir155*-Tg group showed significantly reduced BV/TV and Tb.N in newly formed bone compared with the wild-type group (*Figure 4B and C*). Tb.Th and Tb.Sp levels were similar in the *Mir155*-Tg and wild-type groups (*Figure 4D and E*). In contrast, ectopic bone regeneration was significantly increased in the *Mir155*-KO group compared with the wild-type group (*Figure 4F*). Newly formed bone BV/TV and Tb.N in the *Mir155*-KO group were increased by 6.12- and 5.64-fold respectively compared with the wild-type group (*Figure 4G and H*). Tb.Th remained unchanged in *Mir155*-KO mice (*Figure 4I*) but the Tb.Sp was significantly reduced in the *Mir155*-KO group compared with the wild-type group (*Figure 4J*). These results indicate a catabolic effect of *Mir155* on bone regeneration.

### *Mir155*-KO mice showed enhanced bone regeneration in an orthotopic model

We further added a low dose of BMP-2 to the collagen membrane and evaluated the bone regeneration in calvarial bone defect of *Mir155*-KO and wild-type mice. The defect area was covered with

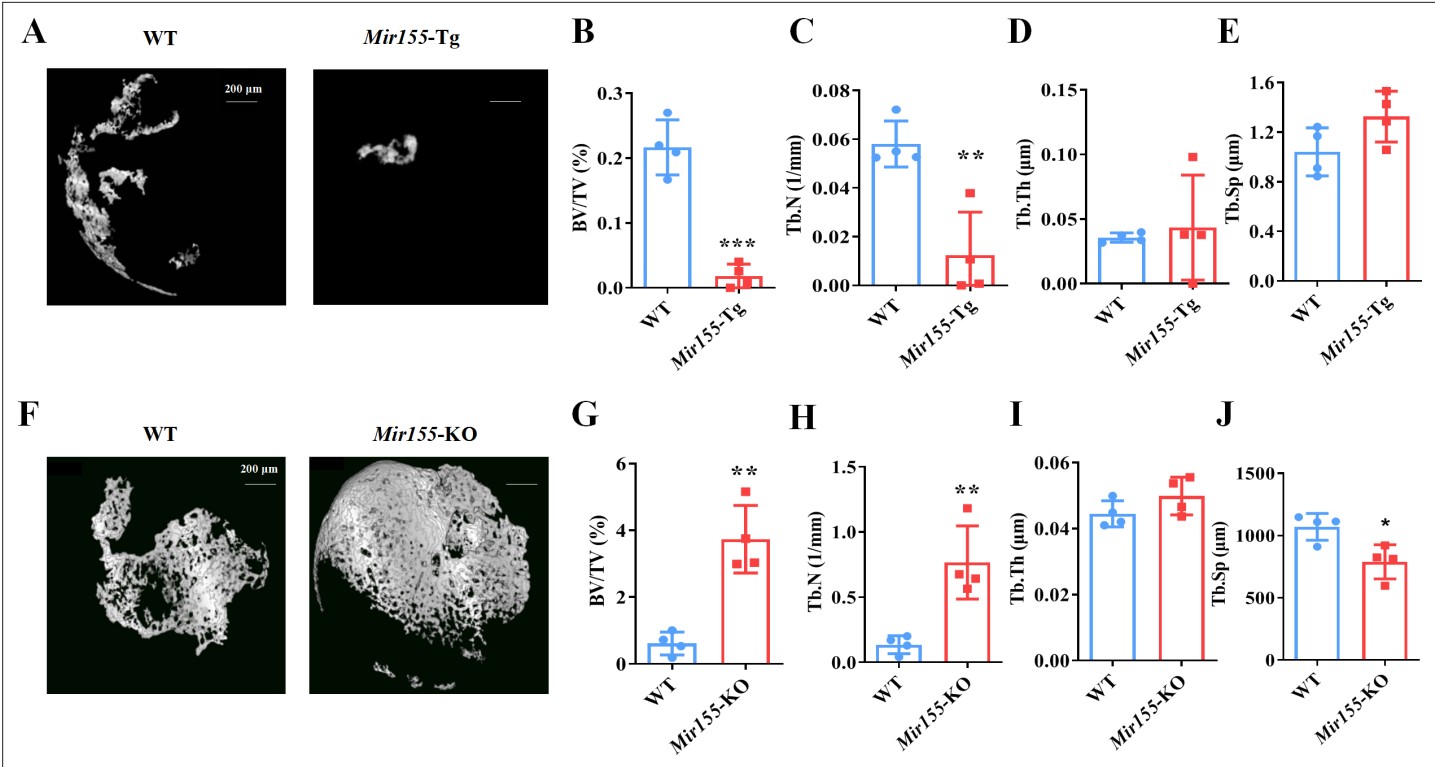

**Figure 4.** Ectopic bone regeneration was inhibited in *Mir155* transgenic (*Mir155*-Tg) mice but enhanced in *Mir155* knockout (*Mir155*-KO) mice. (**A**) Representative micro-CT images. (**B**) Bone volume/total volume (BV/TV), (**C**) trabecular number (Tb.N), (**D**) trabecular thickness (Tb.Th), and (**E**) trabecular separation (Tb.Sp) analysis in *Mir155*-Tg and wild-type mice. (**F**) Representative micro-CT images. (**G**) BV/TV, (**H**) Tb.N, (**I**) Tb.Th, and (**J**) Tb. Sp analysis in *Mir155*-KO and wild-type mice. Data are presented as mean ± SD, n=4. Significant difference compared to wild-type group, *p<0.05, **p<0.01, and ***p<0.001.

The online version of this article includes the following source data for figure 4:

**Source data 1.** Raw data for *Figure 4B, E, G-J*.

a robustly high amount of newly formed bone in *Mir155*-KO mice compared with wild-type mice (*Figure 5A and B*). Furthermore, BV/TV, BMD, and Tb.N (*Figure 5C–E*) were enhanced while Tb.Th and Tb.Sp were reduced in *Mir155*-KO mice compared with wild-type mice (*Figure 5F and G*). These results from calvarial bone defect healing analysis showed promising bone regeneration effects of *Mir155*-KO.

### *Mir155* influences the osteogenic differentiation of BMSCs

To further confirm the regulatory role of *Mir155* on osteogenesis, we analyzed the osteogenic differentiation potential BMSCs isolated from *Mir155*-Tg, *Mir155*-KO, and the respective wild-type mice. Mineralized matrix deposition potential in BMSCs from *Mir155*-TG mice was substantially reduced compared with that of wild-type mice (*Figure 6A and B*). Similarly, protein expression levels of osteogenic markers ALP and RUNX2 in BMSCs from *Mir155*-Tg mice were reduced compared with those of wild-type mice (*Figure 6C and D*). These results indicate the compromised osteogenic differentiation potential of BMSCs from *Mir155*-Tg mice. In contrast, BMSCs from *Mir155*-KO mice showed robustly higher matrix mineralization potential compared to those of wild-type mice (*Figure 6E and F*). The protein expression levels of osteogenic markers ALP were enhanced in BMSCs from *Mir155*-KO (*Figure 6G and H*). However, RUNX2 protein levels were not changed in BMSCs from *Mir155*-KO mice compared with those from wild-type mice. These results demonstrated the catabolic effect of *Mir155* in the osteogenic differentiation of BMSCs.

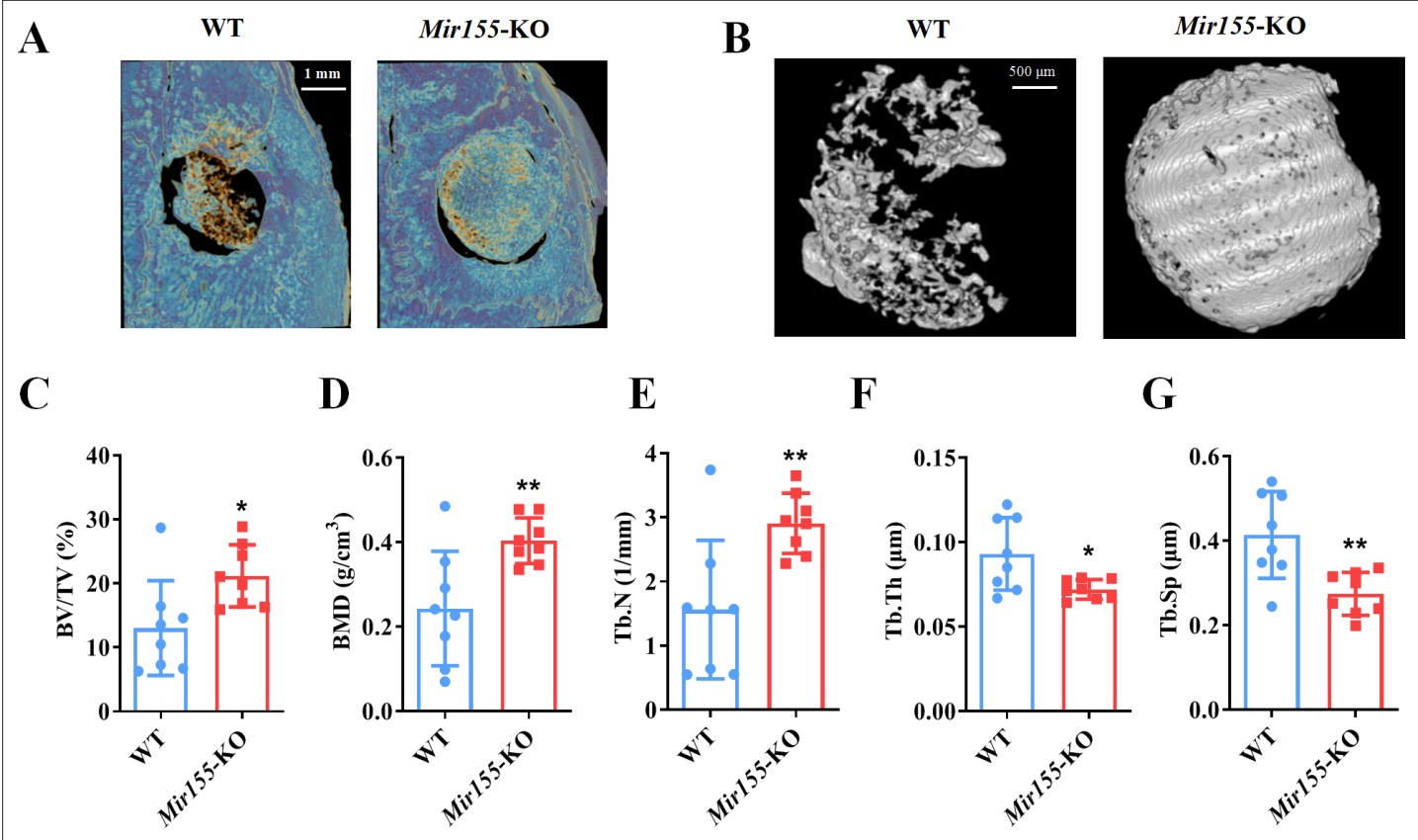

**Figure 5.** A higher degree of bone regeneration was observed in the calvarial defect of *Mir155* knockout (*Mir155*-KO) mice with a low dose of BMP2 treatment. (**A**) Representative micro-CT images, (**B**) local micro-CT images in defects, (**C**) bone volume/total volume (BV/TV), (**D**) bone mineral density (BMD), (**E**) trabecular number (Tb.N), (**F**) trabecular thickness (Tb.Th), and (**G**) trabecular separation (Tb.Sp) analysis. Data are presented as mean ± SD, n=8. Significant difference compared to wild-type mice, *$p<0.05$ and **$p<0.01$.

The online version of this article includes the following source data for figure 5:

**Source data 1.** Raw data for *Figure 5C-G*.

### *Mir155* knockdown promotes the osteogenic differentiation of BMSCs

*Mir155* knockdown in BMSCs was used to further confirm the role of *Mir155* in osteogenic differentiation. *Mir155* sponge lentivirus treatment significantly reduced the expression of *Mir155* in BMSCs (*Figure 7A*), indicating the successful knockdown of *Mir155*. Matrix mineralization was robustly increased in the *Mir155* sponge group compared with the negative control group (*Figure 7B and C*). Furthermore, similarly, the protein expression levels of osteogenic markers ALP and RUNX2 were robustly upregulated in the *Mir155* sponge group compared with the negative control group (*Figure 7D and E*). Furthermore, sponging *Mir155* did not affect cell viability and expression of mesenchymal stem cell markers in BMSCs (*Figure 7F and G*). These results further confirm the catabolic effect of *Mir155* on the osteogenic differentiation of precursor cells.

### *Mir155* targets the *S1pr1* gene to regulate the osteogenic differentiation of BMSCs

TargetScan prediction showed that the binding sites of *Mir155* on *S1pr1* were rather conserved in different species, such as human, mice, rat, rhesus, etc. (*Figure 8A*). The sequences of seeding sites and mutant seeding sites of *S1pr1* are shown in *Figure 8B*. Luciferase reporter gene assay was performed to analyze the *Mir155* and *S1pr1* gene interaction (*Figure 8C*). Our results showed that *Mir155* directly binds to the 3'UTR of the S1PR1 (*Figure 8C*). Sponging *Mir155* robustly enhanced the protein level expression of the *S1pr1* gene in BMSCs (*Figure 8D*), confirming the interaction of *Mir155* and the *S1pr1* gene. *S1pr1* transfection in BMSCs robustly enhanced the matrix mineralization

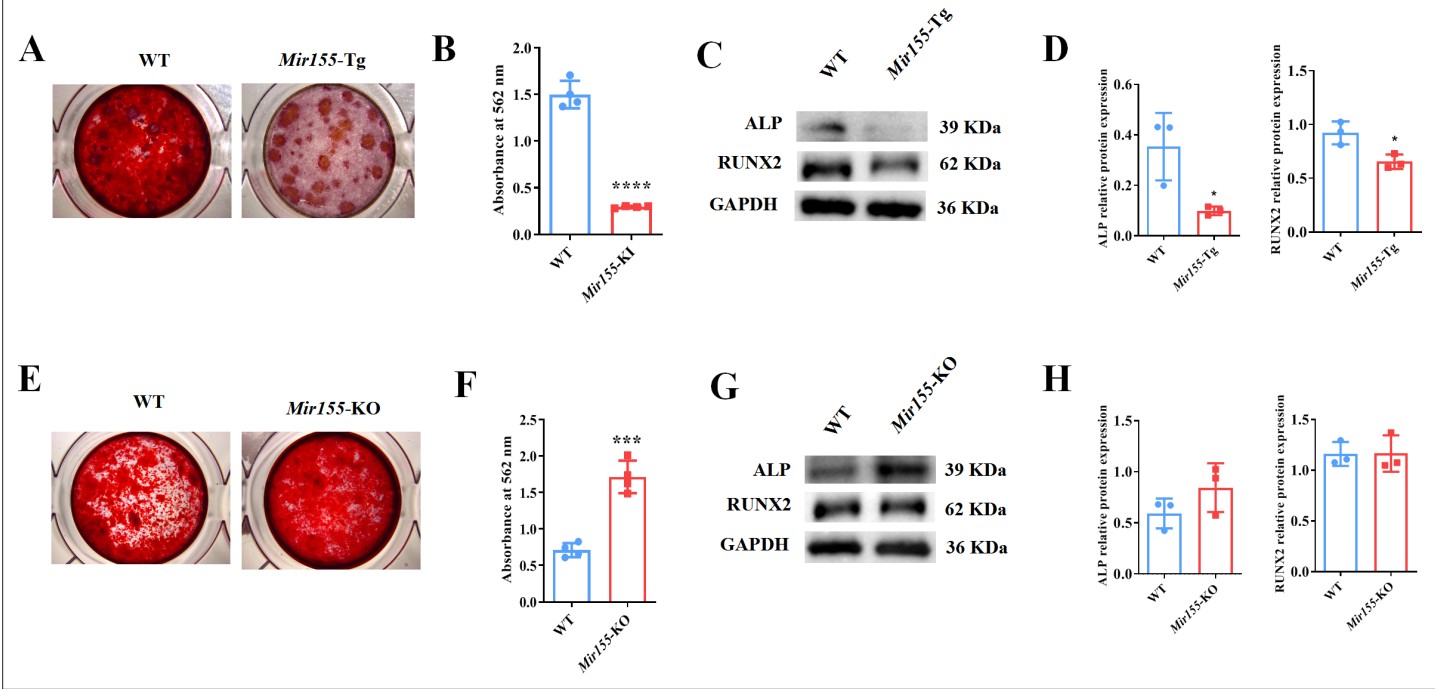

**Figure 6.** *Mir155* transgenic (*Mir155*-Tg) and *Mir155* knockout (*Mir155*-KO) bone marrow stromal stem cells (BMSCs) showed an opposite trend of osteogenic differentiation. (**A**) Alizarin red staining (ARS) images at day 10 of culture, (**B**) ARS quantification, n=4, (**C**) Western blot analysis of osteogenic markers, and (**D**) densitometry quantification of protein bands in Mir155-Tg BMSCs, n=3. (**E**) ARS images stained at day 10 of culture, (**F**) ARS quantification, n=4, (**G**) Western blot analysis of osteogenic markers, and (**H**) densitometry quantification of protein bands in *Mir155*-KO BMSCs, n=4. Data are presented as mean ± SD. Significant difference compared to wild-type mice, *p<0.05, ***p<0.001, and ****p<0.0001.

The online version of this article includes the following source data for figure 6:

**Source data 1.** Raw data for **Figure 6B, D** (ALP and RUNX2), **Figure 6E, H** (ALP and RUNX2); original blots for **Figure 6C and G**.

(**Figure 8E and F**). The protein expression level of S1PR1 was enhanced in lentivirus-mediated *S1pr1* overexpressed BMSCs (**Figure 8G and H**). This result indicates the efficacy of lentivirus-based *S1pr1* overexpression in BMSCs. The protein expression levels of ALP and RUNX2 were increased in *S1pr1* overexpressed BMSCs (**Figure 8G and H**). *S1pr1* overexpression in BMSCs by lentivirus did not affect cell viability and proliferation (**Figure 8I**). These results indicate that the *Mir155* targets the *S1pr1* gene to regulate the osteogenic differentiation of BMSCs.

### *Mir155* influences osteoclastogenesis

The function of *Mir155* on osteoclastogenesis was further explored. Primary bone marrow monocytes were isolated and induced into bone marrow-derived macrophages (BMMs). The receptor activator of nuclear factor-κB (NF-κB) ligand (RANKL) was further used for osteoclastogenic differentiation. Tartrate-resistant acid phosphatase (TRAP) staining showed that *Mir155*-KO reduced the osteoclast number (**Figure 9A and B**). Gene expression of osteoclastogenesis markers including receptor activator of NF-κB (*Rank*) and cathepsin K (*Ctsk*) was significantly reduced during the osteoclastogenic differentiation of *Mir155*-KO BMMs (**Figure 9C**). Furthermore, bone resorption-related C-terminal telopeptides of type I collagen (CTX-1) level were significantly reduced while bone formation-related procollagen type I N-terminal pro-peptide (PINP) level was increased in serum of *Mir155*-KO mice compared with wild-type mice (**Figure 9D**). These results demonstrate that *Mir155*-KO has the potential inhibitory effect on osteoclastogenesis.

## Discussion

Differentiation of MSCs to osteoblasts is a vital event of bone regeneration. Differentiated osteoblasts deposit mineralized matrix and contribute to new bone formation. Various miRNAs have been reported to regulate osteogenesis and bone mass phenotype (**Arriaga et al., 2019**). In this study,

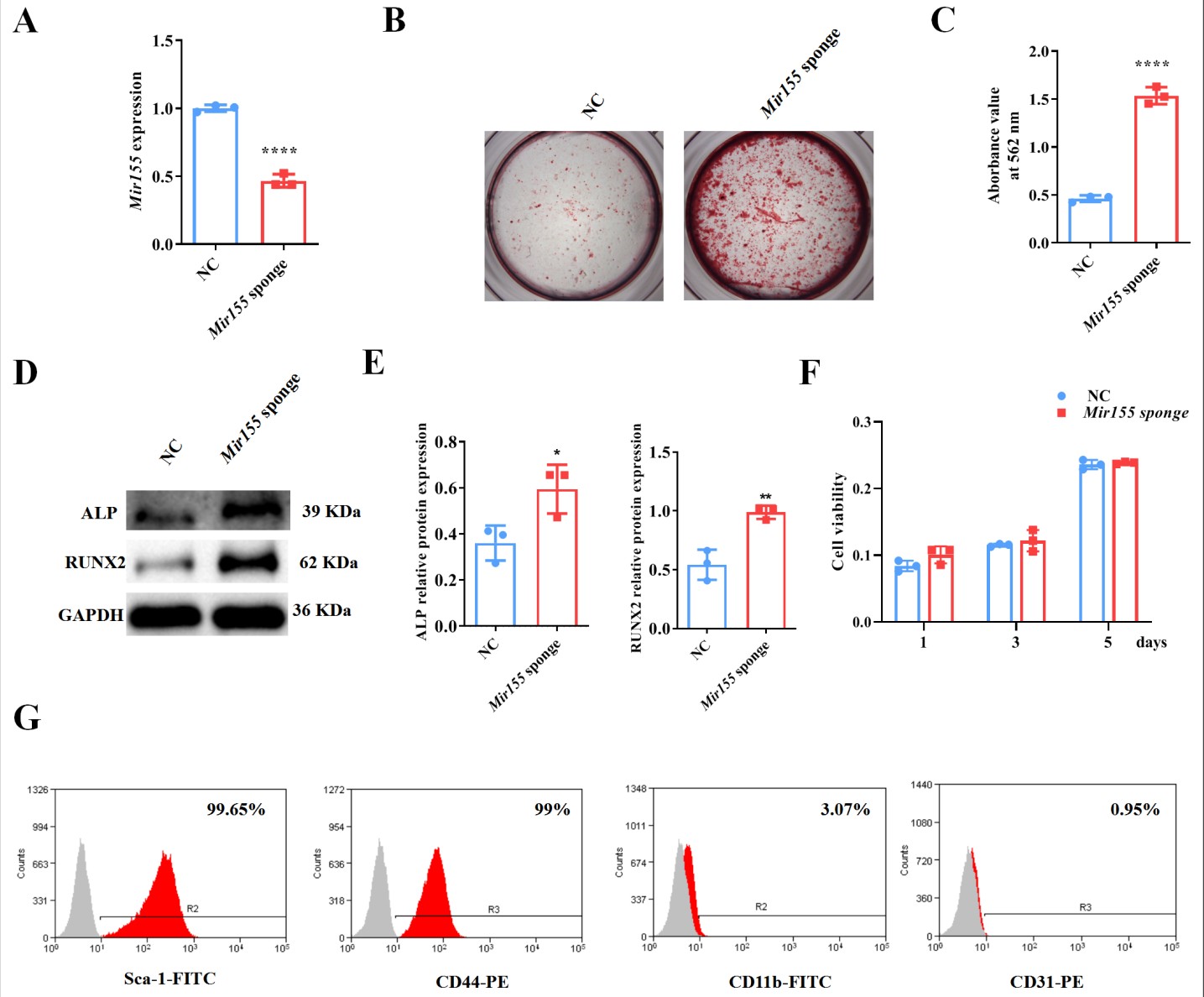

**Figure 7.** *Mir155* knockdown bone marrow stromal stem cells (BMSCs) showed higher osteogenic differentiation potential. (**A**) Mir155 expression level, n=3, (**B**) Alizarin red staining (ARS) images stained at 21 days of culture, (**C**) ARS quantification, n=3, (**D**) Western blot analysis, (**E**) densitometry quantification of protein bands, n=3, (**F**) cell viability, n=3, and (**G**) Fluorescence activated cell sorting (FACS) analysis. Data are presented as mean ± SD. Significant difference compared to the negative control, *p<0.05, **p<0.01, and ****p<0.0001. NC: negative control.

The online version of this article includes the following source data for figure 7:

**Source data 1.** Raw data for *Figure 7A, C, E* (ALP and RUNX2), and *Figure 7F*; original blots for *Figure 7D*.

*Mir155*-Tg mice showed compromised bone regeneration and low bone mass phenotype. In contrast, *Mir155*-KO mice showed improved bone regeneration, higher bone mass phenotype, and a protective effect against inflammation-induced bone loss. BMSCs from *Mir155*-Tg and *Mir155*-KO mice showed compromised and robust osteogenic differentiation potential, respectively. *Mir155* knockdown also promoted osteogenic differentiation potential in BMSCs. These results indicate a catabolic effect of *Mir155* on bone regeneration and bone mass phenotype. Knockdown of *Mir155* in BMSCs robustly enhanced the protein level expression of S1PR1 and osteogenic regulator RUNX2, indicating S1PR1 as a target gene of *Mir155* in BMSCs to regulate osteogenic differentiation. *S1pr1* overexpression in BMSCs enhanced RUNX2 expression and osteogenic differentiation of BMSCs indicating the regulatory role of *Mir155*-S1PR1 interaction on osteogenesis (*Figure 10*).

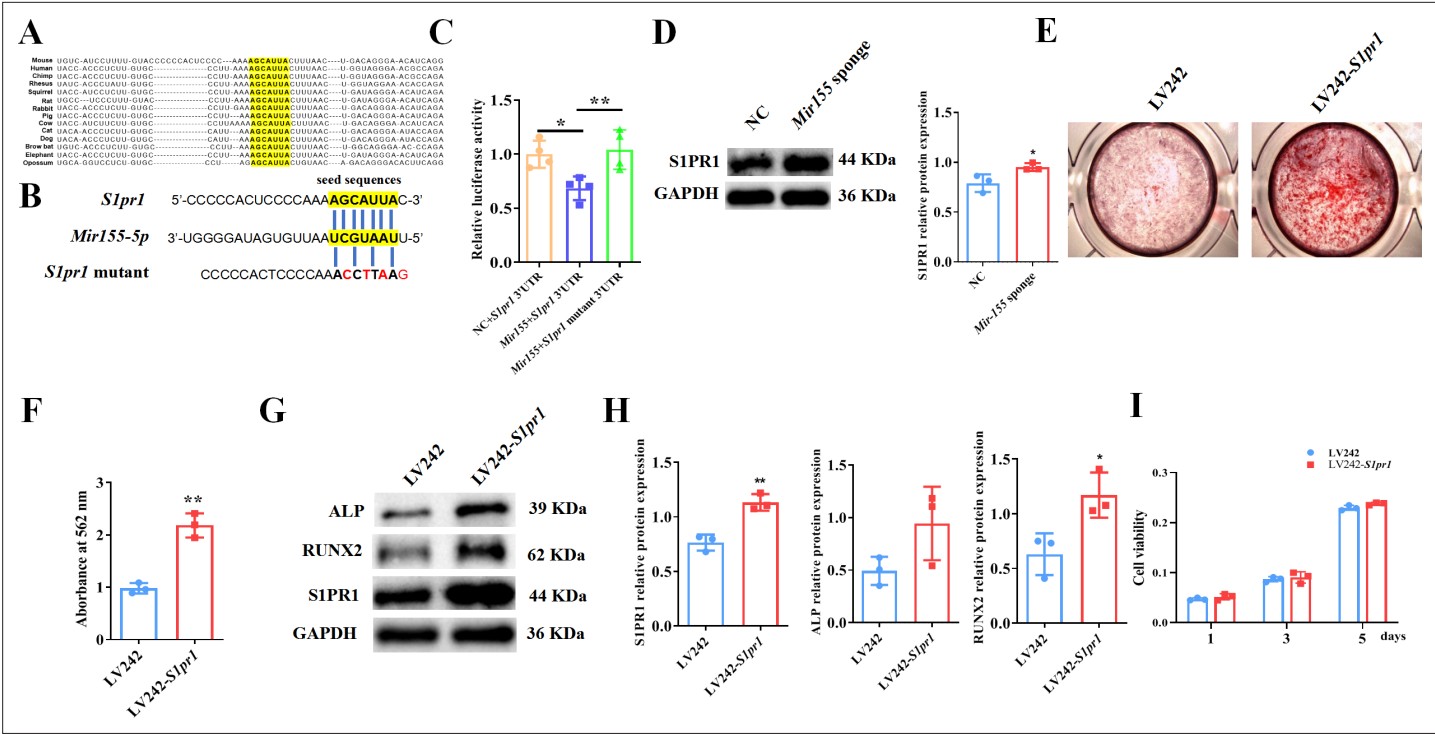

**Figure 8.** *Mir155* targets sphingosine 1-phosphate receptor-1 (S1PR1) to regulate the osteogenic differentiation of bone marrow stromal stem cells (BMSCs). (**A**) The *Mir155* binding site of *S1pr1* in different species, (**B**) the wild and mutant binding site of *S1pr1* in mice. (**C**) Luciferase assay, n=4, (**D**) S1PR1 protein expression and densitometry quantification of protein bands, n=3, (**E**) Alizarin red staining (ARS) images stained at 21 days of culture, (**F**) ARS quantification, n=3, (**G**) Western blot analysis of osteogenic markers, (**H**) densitometry quantification of protein bands, n=3, and (**I**) cell viability analysis, n=4. Data are presented as mean ± SD. Significant difference compared to negative control, *p<0.05 and **p<0.01.

The online version of this article includes the following source data for figure 8:

**Source data 1.** Raw data for *Figure 8C, D, F, H and I*; original blots for *Figure 8D, G*.

miRNAs/anti-miRNAs have been used for bone tissue engineering (*Arriaga et al., 2019*). MiR26a (*Wang et al., 2015*), anti-miR31 (*Deng et al., 2013*), anti-miR34a (*Chen et al., 2020b*), miR135 (*Xie et al., 2016*), anti-miR138 (*Eskildsen et al., 2011*), anti-miR146a (*Xie et al., 2017*), miR148a (*Li et al., 2017*), anti-miR221 (*Yeh et al., 2016*), and anti-miR3555p *Tomé et al., 2011* have shown an anabolic effect on osteogenic differentiation of precursor cells and bone regeneration. Most of the miRNAs/anti-miRNAs promote bone regeneration via the activation of the osteogenic master regulator RUNX2 (*Arriaga et al., 2019*). In this study, *Mir155* overexpression showed a catabolic effect on osteogenesis and bone mass phenotype. Interestingly, knockout or knockdown of *Mir155* showed an anabolic effect on osteogenesis, bone regeneration, and bone defect healing. RUNX2 was involved in the *Mir155*-mediated regulation of osteogenic differentiation of BMSCs. RUNX2 expression did not change in *Mir155*-KO BMSCs but was upregulated in *Mir155* sponged BMSCs from wild-type mice. Whole body knockout of *Mir155* also knock outs the *Mir155* in the other cell types present in bone microenvironment, for example, monocytes and macrophages. The *Mir155*-KO BMSCs might have certain effects of surrounding *Mir155*-KO other cell types on S1PR1 expression, while the *Mir155* sponged BMSCs from wild-type mice sorted with FACS may be free from such effects. This could be the possible explanation for the lack of change in RUNX2 expression in *Mir155*-KO BMSCs, but rather an upregulation of RUNX2 expression in *Mir155* sponged BMSCs from wild-type mice observed in this study. However, further studies are required to support this logic. *Mir155* had been reported to inhibit osteogenesis of MC3T3-E1 cells via SMAD5 downregulation (*Gu et al., 2017*). *Mir155* inhibits BMP9-induced osteogenic differentiation of precursor cells via downregulation of BMP signaling (*Liu et al., 2018*). Qu et al. revealed that miR155 inhibition alleviates high glucose and free fatty acid-suppressed osteogenic differentiation of BMSCs by targeting SIRT1 (*Qu et al., 2020*). Our results from *Mir155*-Tg mice on inhibition of osteogenesis are in accordance with the findings from the literature (*Gu et al., 2017;*

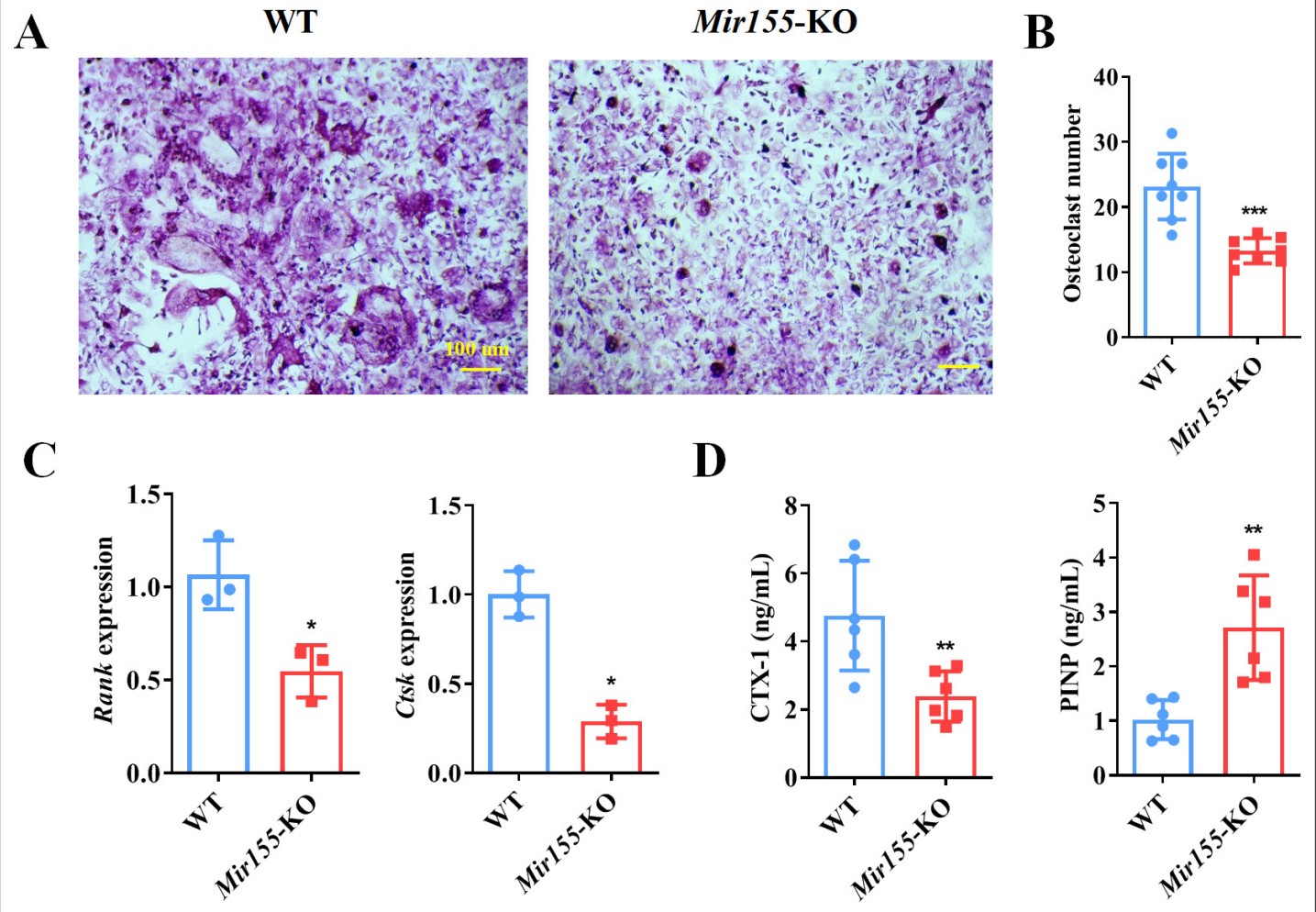

**Figure 9.** Bone marrow-derived macrophages (BMMs) from *Mir155* knockout (*Mir155*-KO) mice exhibited compromised osteoclastogenic differentiation. (**A**) Tartrate-resistant acid phosphatase (TRAP) staining, (**B**) osteoclast number quantification, n=8, (**C**) osteoclastogenic markers expression, n=3, (**D**) C-terminal telopeptides of type I collagen (CTX-1), and procollagen type I N-terminal pro-peptide (PINP) serum level, n=6. Data are presented as mean ± SD. Significant difference compared to the negative control, *p<0.05, **p<0.01, and ***p<0.001.

The online version of this article includes the following source data for figure 9:

**Source data 1.** Raw data for *Figure 9B, C and D*.

*Liu et al., 2018*). Furthermore, *Mir155*-KO inhibited the osteoclastogenic differentiation of BMMs. To our knowledge, this is the first study to report the anabolic effect of *Mir155*-KO or knockdown on osteogenic differentiation and the catabolic effect of *Mir155*-KO on osteoclastogenesis. Our results suggest the potential application of anti-miR155 on bone regeneration and bone tissue engineering applications.

Anti-miR155 oligonucleotides and antagomir have been designed for various cancer treatments (*Kardani et al., 2020*; *Witten and Slack, 2020*). Small molecule-based cyclic peptidomimetics had shown an inhibitory effect on miR155 biogenesis (*Yan et al., 2019*). MLN4924 is an inhibitor of the NEDD8-activating enzyme. MLN4924 decreases the binding of NF-κB to the miR155 promoter and downregulates miR155 in AML cells (*Khalife et al., 2015*). Since the knockout of *Mir155* promoted bone regeneration and protected against inflammation-induced bone loss, anti-Mir155 or Mir155 inhibitors could be applied for bone tissue engineering and the treatment of low bone mass pheno-type and inflammation-related bone loss. However, the osteoinductive potential of already available anti-*Mir155* or *Mir155* inhibitors should be tested using in vitro and in vivo models to prove this hypothesis.

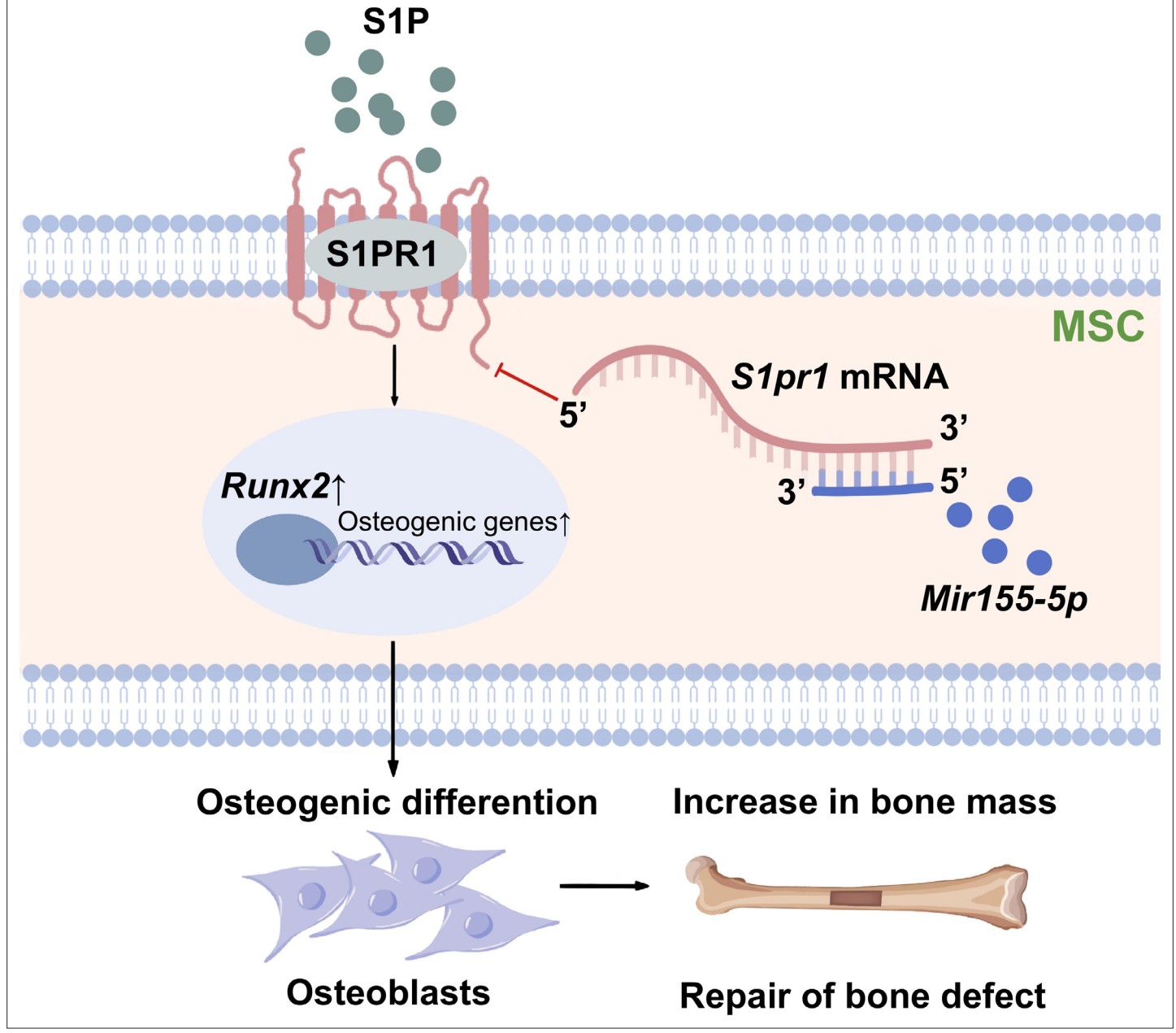

**Figure 10.** Scheme of *Mir155*-mediated regulation of osteogenesis. S1P activates the sphingosine 1-phosphate receptor-1 (S1PR1), further increasing RUNX2 expression to regulate the osteogenic differentiation of MSCs into osteoblasts. *Mir155* inhibits this process by direct binding with 3'UTR *S1pr1* mRNAs. MSCs: mesenchymal stromal cells.

miR155 targets different genes in different cells to regulate the cell type-specific functions (*Woeller et al., 2019*; *Wang et al., 2018*; *Hawez et al., 2019*; *Li et al., 2021c*). *S1pr1*, a target gene of *Mir155*, is regulated during various physiological and pathological conditions (*Li et al., 2021c*; *Okoye et al., 2014*). TargetScan prediction results showed that the binding site of *Mir155* on *S1pr1* was rather conserved in several different species such as human, rat, mouse, etc. In this study, *Mir155* inhibition upregulated S1PR1 protein expression. Overexpression of *S1pr1* robustly promoted RUNX2 protein expression and osteogenic differentiation of BMSCs. *Mir155* has been shown to inhibit the osteogenic differentiation of precursor cells via inhibiting SMAD5 (*Gu et al., 2017*). Higashi et al. reported SMAD1/5/8 as downstream signaling of S1PR1/S1PR2 to induce RUNX2 expression in osteoblasts (*Higashi et al., 2016*). Reports from the literature and results of this study indicate that *Mir155* targets the *S1pr1* gene to inhibit RUNX2 expression thereby reducing bone regeneration and bone mass.

Since miR155 is upregulated in various cancers including hematological cancers (*Witten and Slack, 2020*). Hematological cancer mainly affects bone marrow which is the dwelling of bone precursor cells. Hematological cancers are associated with bone loss and fracture of vertebrae and long bones. Breast and lung cancer are frequently metastasized to bone and cause osteolysis (*Pang et al., 2020*; *Cheng et al., 2021*). Cancer/cancer metastasis-induced bone loss-mediated fracture is a serious clinical problem. However, the role of upregulated levels of miR155 on cancer/cancer metastasis-related reduced bone mass is still unclear. Moreover, the prevention of cancer/cancer metastasis-induced bone loss is a huge challenge for clinicians. Since anti-miR155 has already been proven to be beneficial for cancer treatment and miR155 knockdown promotes bone regeneration, anti-miR155 could treat cancer as well as caner-induced bone loss as a killing two birds with one stone concept. However, future in vitro and in vivo studies are needed to confirm this hypothesis.

Mir155 is overexpressed and plays a key role in the pathophysiology of inflammatory diseases including autoimmune arthritis, osteoarthritis, and periodontitis (*Wu et al., 2021*; *Blüml et al., 2011*; *Li et al., 2021a*). An elevated level of *Mir155* in arthritis promotes M1 macrophage polarization and inflammation (*Li et al., 2021a*). Prevention of bone loss in inflammatory diseases using currently available therapeutic approaches is not satisfactory. Moreover, inflammation impedes bone regeneration thereby causing the failure of bone tissue engineering approaches. In the present study, we found that *Mir155*-KO exerts a protective effect against LPS-induced bone loss. Since anti-*Mir155* has anti-inflammatory (*Teng et al., 2020*) and bone regenerative potential, anti-*Mir155* could be a potential therapeutic to prevent inflammation-induced bone loss.

This study used both *Mir155*-Tg and *Mir155*-KO mice to investigate the role of *Mir155* on osteogenesis and bone mass phenotype. Bone regeneration in both ectopic and orthotopic models confirmed the regulatory role of *Mir155* in bone regeneration. *Mir155* silenced and *S1pr1* overexpressed BMSCs further confirmed the *S1pr1* as a target gene of *Mir155* to regulate RUNX2 expression during osteogenesis. The limitation of this study is that we did not analyze the downstream signaling pathway of S1PR1 that regulates RUNX2 expression in BMSCs. Clinical application of osteogenic factors in vivo always poses the risk of vascular calcification. miR155-5p overexpression has been reported to aggravate vascular calcification (*He et al., 2020*). Importantly, *Mir155*-KO mice also show resistance against vitamin D3-induced vascular calcification (*Li et al., 2021b*). Moreover, *Mir155* deletion inhibits the migration and apoptosis of vascular smooth muscle cells as well as vascular calcification (*Li et al., 2021b*). Furthermore, Zhang et al. showed normal histology of vital organs including the heart, lung, liver, and spleen in *Mir155*-KO mice (*Zhang et al., 2017*). These reports from the literature indicate that *Mir155*-KO does not pose the risk of vascular calcification. However, the effect of *Mir155* inhibitors or anti-*Mir155* on vascular calcification should be thoroughly investigated before applying these agents for bone regeneration applications.

In conclusion, *Mir155* showed a catabolic effect on osteogenesis and bone mass via targeting *S1pr1*. Our results suggest miR155 as a potential target to promote bone regeneration and higher bone mass. Since miR155 is overexpressed in inflammatory diseases and anti-miR155 has shown anti-inflammatory potential, the miR155 inhibitors could be the potential therapeutics to promote bone regeneration even in inflammatory conditions.

## Materials and methods

### Mice

*Mir155*-KO mice were purchased from the Jackson Laboratory (Stock No. 007745). *Mir155*-Tg mice were constructed as described in our previous reports (*Lin et al., 2016*). The C57BL/6J wild-type mice, as the wild-type mice of *Mir155*-KO mice, were purchased from Guangdong Medical Laboratory Animal Center. While the FVB mice were a littermate control of *Mir155*-Tg mice. The blinded evaluation was used for mice assignments and analysis. The animal experiment was conducted in accordance with the guidelines approved by the Institutional Animal Care and Use Committee of the First Affiliated Hospital of Guangzhou Medical University, Guangzhou, China (2017-078).

### Bone phenotype analysis

Bone phenotype analysis was performed in 8-week-old mice (8 mice/group including 4 male and 4 female mice) using micro-CT. Mice were anesthetized using isoflurane (RWD Life Science Co., China),

followed by cervical dislocation. The femur with a distal growth plate was collected and fixed in 10% buffered formalin. Micro-CT scanning was performed to evaluate bone phenotype using Bruker Sky1172 Skyscan (Kontich, Belgium). A total of 100 slices (1 mm) below the distal growth plate of the femurs was measured for 3D reconstruction and quantification of trabecular bone and cortical bone as described previously (*Moussa et al., 2021*). The X-ray tube was operated at 96 kV and 65 μA using a 0.5 mm Al filter with a resolution of 7.93 μm pixels. Scanning was performed by 180° rotation around the vertical axis, camera exposure time of 1300 ms, rotation step of 0.6°, frame averaging of 2, and random movement of 10. 3D images were made using CTvox software (Skyscan, Kontich, Belgium). Data viewer software (Skyscan, Kontich, Belgium) was used for images and linear analysis. Relative bone formation parameters including BV/TV, BMD, Tb.N, Tb.Th, and Tb.Sp were analyzed.

### LPS-treated mice

As reported by *Chen et al., 2020a*, wild-type mice and Mir155-KO mice (6 mice/group including 3 male mice and 3 female mice) were injected intraperitoneally with LPS (8 mg/kg) twice 1 week. Mice were weighed before LPS injection. After 6 weeks, femurs were collected for micro-CT analysis and H&E staining.

### H&E staining

H&E staining for bone tissues was performed as previously reported (*Chen et al., 2020a*; *Chen et al., 2018*). Femurs were fixed in 4% PFA for 2 days and decalcified with EDTA decalcified solution for 21 days. After that, femurs were cut for histological analysis, the femurs were dissected and conducted in 4% PFA for 2 days. Then slices of bone tissue at 3 μm thickness were cut along the coronal plate. The decalcified slices were further performed H&E staining by dewaxing, hematoxylin staining, eosin staining, and dehydration.

### Ectopic grafting of collagen membrane

The subcutaneous transplantation of the collagen membrane was performed as described previously (*Huang et al., 2017*). Collagen membrane ZH-BIO (China) with 5 mm diameter and 1 mm thickness were osteogenically functionalized by loading 10 μL of 0.3 mg/mL BMP2 solution. BMP2-loaded collagen membranes were implanted in the subcutaneous pockets of *Mir155*-Tg, *Mir155*-KO, and respective wild-type mice. Eight-week-old male mice with 20–22 g body weight (4 mice/group, 1 membrane/mouse) were used for this study. After 18 days of transplantation, mice were euthanized by isoflurane, collagen membrane was collected and further analyzed for newly formed bone using micro-CT.

### Mice calvaria bone defect healing with BMP2 addition

As previously reported (*Reyes et al., 2018*), a 3 mm diameter was generated on one side of the sagittal suture in 8-week *Mir155*-KO mice and wild-type mice (8 mice/group including 4 male mice and 4 female mice, 1 defect/mouse). A total of 200 ng BMP2 (10 μL) was dropped into the collagen membrane ZH-BIO (China). The membrane was inserted into the defect. Calvaria bones were collected and analyzed by micro-CT after 1 month.

### The isolation of primary BMSCs and osteoclastogenic induction

Euthanized transgenic mice and wild-type male mice (5–6 weeks of age) were immersed into 75% ethanol for 5 min. The femurs and tibia were acquired. Primary BMSCs were isolated and expanded as described previously (*Soleimani and Nadri, 2009*). In brief, bone marrow was flushed out from the tibia and femurs and disturbed into small pieces. Cells were collected by centrifugation and plated into flasks and allowed to adhere for 24 hr. Nonadherent cells were washed, and culture was continued in DMEM supplemented with 10% non-heat inactivated FBS and 1% penicillin/streptomycin. The cells were cultured in a 5% $CO_2$ incubator maintaining a humid atmosphere. Cells were trypsinized from 80% confluent culture and passaged.

### Analysis of the target gene of *Mir155*

TargetScan software was used to predict the target gene of *Mir155*. TargetScan predicted S1PR1 as a possible target gene of *Mir155*.

## Plasmid construction and lentivirus preparation

The *S1pr1* 3'UTR sequences and mutant sequences (200 bp upstream and 200 bp downstream of the binding site from NM_007901.5 transcript) were synthesized and cloned into wild-type plasmid pmirGLO Dual-Luciferase miRNA Target Expression Vector (Promega, USA) by Generay (China). We created *S1pr1* 3'UTR and *S1pr1* mutant 3'UTR plasmids for luciferase assay. The LV242 *S1pr1* and control LV242 plasmid were purchased from Genecopedia (USA). *Mir155* negative control (NC) and *Mir155* sponge plasmids were purchased from OBIO (China).

As previously described (*Zheng et al., 2013*), HEK293T cells were co-transfected with expression plasmids (*Mir155* NC, *Mir155* sponge, LV242, or LV242 *S1pr1*) with the packaging plasmids pMD2. VSVG, pMDLg/pRRE, and pRSV-REV using EZ trans transfection regent (Shanghai life iLab BioTechnology Co., Ltd., China). After 48 hr, fresh lentiviral supernatant was collected and used for infection. BMSCs were expanded to 60% confluence before lentiviral infection. After infection for 10 hr, cells were washed and allowed to recover for 24 hr and used for subsequent experiments. *Mir155* sponge efficacy was analyzed by RT-qPCR. S1PR1 overexpression efficacy was analyzed by Western blot analysis.

## Luciferase assay

Luciferase assay was performed as previously described (*Wang et al., 2019c*). Luciferase assay was performed for further confirmation of *S1pr1* as a target gene of *Mir155*. S1PR1 3'UTR (100 ng) with NC (50 nM), *S1pr1* 3'UTR (100 ng) with *Mir155* (50 nM), and the *S1pr1* mutant 3'UTR plasmid (100 ng) with *Mir155* (50 nM) were co-transfected into HEK293T cells by Lipofectamine 2000 (Thermo Fisher Scientific Inc, USA). After 48 hr, the luciferase assay was performed according to the manufacturer's instructions using Luc-Pair Duo-Luciferase Assay Kit 2.0 (GeneCopoeia, USA). In brief, the cells were lysed and incubated with 100 µL Fluc work solution for 5 min, the fluorometric measurement was performed by Varioskan Flash (Thermo Fisher, USA). The lysis solution was added with 100 µL Rluc, incubated for 5 min, and the fluorometric value was measured.

## Alizarin red staining

Alizarin red staining was performed as a previous report (*Wang et al., 2022*). BMSCs (28,000 cells/well) were seeded at 48-well culture plates and cultured with osteogenic medium (50 µg/mL vitamin C, 0.01 µM dexamethasone, and 10 mM β-glycerophosphate). Then, the cells were fixed with paraformaldehyde and stained with Alizarin Red S solution (1%, pH 4.2) (Solarbio Life Science, China) for 10 min. The staining was visualized under stereomicroscope Leica EZ4HD (Leica, Germany). For quantitative analysis, the alizarin red-stained mineralized matrix was dissolved with 200 µL 10% hexadecylpyridinium chloride monohydrate for 1 hr, and the supernatant was collected. The optical density of the supernatant (100 µL) was measured by a microplate reader at 562 nm.

## Immunoblotting

Immunoblotting was performed as a previous report (*Zhang et al., 2021*). Cells were lysed by using RIPA Buffer (CWBio, China) containing protease inhibitors to extract total protein. Total protein (20 µg) was added to 10% SDS-polyacrylamide gel. The protein was transferred to PVDF membranes (Millipore, USA) after electrophoresis and blocked for 1 hr with a blocking buffer (Beyotime, China). Then PVDF membranes were incubated with the primary antibodies including ALP (Abcam, UK, 1:3000), RUNX2 (CST, USA, 1:2,000), S1PR1 (Abcam, UK, 1:2,000), and GAPDH (CST, USA, 1:5,000) overnight at 4°C. The membranes were further incubated with horseradish peroxidase-conjugated secondary antibody for 1 hr and reacted with ECL (Millipore, USA). Finally, the photographs were taken by Tanon-5200 system (Tanon, China). The densitometry of protein bands was quantified by ImageJ 1.51K.

## RT-qPCR

The femurs were collected from wild-type, *Mir155*-Tg, and *Mir155*-KO mice. And bones were ground with a high-speed low-temperature tissue homogenizer (Servicebio, China). As previously reported (*Teng et al., 2022*), the miRNAs were extracted from BMSCs and ground bone tissues with the MolPure Cell/Tissue miRNA Kit (Yeasen, China) as per the manufacturer's instructions. The miRNA was further reversed by Tailing reaction using miRNA first strand cDNA synthesis kit (Accurate Biology, China). In brief, 3.75 µL miRNA, 5 µL 2×miRNA RT Reaction Solution, 1.25 µL miRNA RT Enzyme Mix were

**Table 1.** Primers used for RT-qPCR.

| Gene name | Forward sequence (5'->3') | Reversed sequence (5'->3') |
|---|---|---|
| Gapdh | TGTGTCCGTCGTGGATCTG | TTGCTGTTGAAGTCGCAGGA |
| Rank | CCAGGAGAGGCATTATGAGCA | ACTGTCGGAGGTAGGAGTGC |
| Ctsk | FCTCGGCGTTTAATTTGGGAGA | TCGAGAGGGAGGTATTCTGAGT |
| Mir155 | 5'-TAATGCTAATTGTGATAGGGGT-3' | |

incubated at 37°C for 1 hr, 85°C for 5 min. The mRNAs from osteoclasts induced from *Mir155*-KO and wild-type mice were extracted with an RNA extraction kit (Accurate Biology, China) as per the manufacturer's instruction. Total RNA (500 ng) was transcribed with reversed regents (Accurate Biology, China). RT-qPCR was performed using SYBR Green Premix Pro Taq HS qPCR Kit (Accurate Biology, China) on an AriaMx Real-time quantitative PCR machine (Agilent, USA). The PCR conditions were 95°C for 30 s, followed by 40 cycles at 95°C for 5 s and 60°C for 30 s. The fold change relative to the control group was measured by the $2^{-\Delta\Delta Ct}$ method. The primers used were shown in *Table 1*.

## PrestoBlue cell viability assay

Cell viability was analyzed using PrestoBlue cell viability reagent (Thermo Fisher, USA) (*Jaafar et al., 2019*). BMSCs ($4 \times 10^3$ cells/well) were seeded into 96-well culture plates. After 1, 3, and 5 days, the medium was removed and replaced with a cell viability detection medium according to the manufacturer's instructions. After 2 hr, the OD value was measured by a microplate reader at 570 nm with a reference wavelength of 600 nm.

## TRAP staining

TRAP staining was performed as previously reported (*Ho et al., 2017*). For osteoclastogenic differentiation, bone marrow cells were cultured in α-minimum essential medium containing 10% fetal bovine serum and added with M-CSF (10 ng/mL) for 4 days. Cells ($8 \times 10^4$ cells/well) was supplemented with M-CSF (20 ng/mL) and RANKL (25 ng/mL) for 7 days. TRAP staining was performed with Acid Phosphatase, Leukocyte (TRAP) Kit as the manufacturer's introduction (Sigma, USA).

## ELISA

Serum PINP and CTX-1 are essential markers for evaluating the level of bone resorption (*Ye et al., 2021*). The levels of PINP and CTX-1 were measured using the PINP ELISA kit (SAB, USA) and CTX-1 ELISA kit (Finebio, China) as the manufacturer's introduction.

## FACS analysis

FACS analysis was performed as previously reported (*Zhu et al., 2010*). For cell surface staining, Sca-I (1:1000, eBioscience, USA), CD44 (1:1000, eBioscience, USA), CD11b (1:1000, eBioscience, USA), and CD31 (1:200, eBioscience, USA) were used.

## Statistical analysis

Data are expressed as mean ± SD. Statistical analysis was performed with t-tests for the comparison of the two groups. $p < 0.05$ was considered a significant difference.

## Acknowledgements

This project was funded by the Science and Technology program of Guangzhou (202201010073, 202201020116), the National Natural Science Foundation of China (U22A20159, 82150410451), the General Guiding Project of Guangzhou (20201A011105), the Medical Scientific Research Foundation of Guangdong Province (B2020027), the Undergraduate Science and Technology Innovation Project of Guangzhou Medical University (2020A049), and High-level University Construction Funding of Guangzhou Medical University (02-412-B205002-1003017 and 06-410-2106035).

## Additional information

### Funding

| Funder | Grant reference number | Author |
| --- | --- | --- |
| The Science and Techonolgoy program of Guangzhou | 202201010073 | Lihong Wu |
| The Science and Technology program of Guangzhou | 202201020116 | Zhichao Zheng |
| The National Natural Science Foundation of China | U22A20159 | Lihong Wu |
| The National Natural Science Foundation of China | 82150410451 | Janak L Pathak |
| The General Guiding Project of Guangzhou | 20201A011105 | Zhichao Zheng |
| The Medical Scientific Research Foundation of Guangdong Province | B2020027 | Zhichao Zheng |
| The Undergraduate Science and Technology Innovation Project of Guangzhou Medical University | 2020A049 | Ruoshu Tang |
| The High-level University Construction Founding of Guangzhou Medical University | 06-410-2106035 and 02-412-B205002-1003017 | Janak L Pathak |

The funders had no role in study design, data collection and interpretation, or the decision to submit the work for publication.

### Author contributions

Zhichao Zheng, Data curation, Formal analysis, Funding acquisition, Validation, Investigation, Methodology, Writing - original draft; Lihong Wu, Zhicong Li, Formal analysis, Validation, Investigation, Methodology; Ruoshu Tang, Funding acquisition, Investigation, Methodology; Hongtao Li, Formal analysis, Investigation, Methodology; Yinyin Huang, Investigation, Methodology; Tianqi Wang, Shaofen Xu, Formal analysis; Haoyu Cheng, Investigation; Zhitong Ye, Visualization; Dong Xiao, Xiaolin Lin, Resources; Gang Wu, Conceptualization, Supervision, Project administration, Writing – review and editing; Richard T Jaspers, Conceptualization, Data curation, Supervision, Project administration, Writing – review and editing; Janak L Pathak, Conceptualization, Data curation, Supervision, Funding acquisition, Project administration, Writing – review and editing

### Author ORCIDs

Lihong Wu http://orcid.org/0000-0002-4561-9400
Richard T Jaspers http://orcid.org/0000-0002-6951-0952
Janak L Pathak http://orcid.org/0000-0003-2576-443X

### Ethics

The animal experiment was conducted in accordance with the guidelines approved by the Institutional Animal Care and Use Committee of the First Affiliated Hospital of Guangzhou Medical University, Guangzhou, China (2017-078).

### Decision letter and Author response

Decision letter https://doi.org/10.7554/eLife.77742.sa1
Author response https://doi.org/10.7554/eLife.77742.sa2

## Additional files

### Supplementary files
• Transparent reporting form

### Data availability
Source data files have been provided as Figure 1-source data 1, Figure 2-source data 1, Figure 3-source data 1, Figure 4-source data 1, Figure 5-source data 1, Figure 6-source data 1, Figure 7-source data 1, Figure 8-source data 1, Figure 9-source data 1.

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
