## [Editor Report]

The authors have shown the important role of miR155 in promoting bone regeneration and higher bone mass. Their fundamental work shows compelling evidence that miR155 is overexpressed in inflammatory diseases therefore, the use of anti-miR155 could produce anti-inflammatory effects. This shows the importance of miR155 inhibitors as therapeutics to promote bone regeneration in inflammatory conditions.

---

## [Decision Letter]

**Decision letter after peer review:**

Thank you for submitting your article "MicroRNA-155 regulates osteogenesis and bone mass phenotype via targeting S1PR1 gene" for consideration by *eLife*. Your article has been reviewed by 3 peer reviewers, one of whom is a member of our Board of Reviewing Editors, and the evaluation has been overseen by a Reviewing Editor and Mone Zaidi as the Senior Editor. The following individual involved in review of your submission has agreed to reveal their identity: Anirudh Karvande (Reviewer #2).

Essential revisions:

In this study entitled " MicroRNA-155 regulates osteogenesis and bone mass phenotype via targeting S1PR1 gene" the authors have shown that miR155 showed a catabolic effect on osteogenesis and bone mass phenotype via interaction with the Sphingosine 1-phosphate receptor-1 (S1PR1) gene. It is also suggested that inhibition of miR155 is a potential strategy for bone regeneration and bone defect healing. It is an important study as cancer/cancer metastasis-induced bone loss-mediated fracture is a serious clinical problem. The authors have shown that miR155 as a potential strategy for bone regeneration and bone defect healing. However, the authors should address the concerns of the reviewers before the manuscript may be accepted for publication.

1. As cancer/cancer metastasis-induced bone loss is not gender specific it is important to know if the similar effects are observed in female mice? Authors did experiments in 8 week old male mice. Does this high bone mass phenotype in miR-155 KO or miR-155 Tg mice change over the period of time as mice age? It would be interesting in bone field if these mice are resistant/prone to age related bone loss or inflamaging.

2. Bone microarchitecture at the histological level need to be analyzed so as to compare the trabecular integrity in both miR155-Tg and miR155-KO mice. The demonstration of S1PR1 expression in a tissue-specific manner in both the groups through immunohistochemistry would validate the hypothesis in an effective way.

3. The study aims at the osteogenic efficacy of miR155, although experiments involving modulation in the resorption levels will support the results convincingly. TRAP staining for the detection of activated osteoclasts at the cellular level will further validate the hypothesis. Authors should mention and explain these effects in the results and discussion.

4. In figure 3, the authors fail to describe the differences observed among the micro CT parameters of the wild-type group in both studies. Clarification is required in this regard.

5. In figure 4A, authors are advised to represent a good 3D image of calvarial defect where at least the findings are properly depicted. What are the effects on Dynamic histomorphometry in miR-155 TG and miR-155 KO mice? Calcein/Alizarin Red injections followed by histomorphentry will be important to study Bone formation and mineral apposition rates.

6. The findings suggest an overall increase in the bone mass in the absence of miR155, but the experiments are restricted to the trabecular region, authors do not mention any variations observed in the cortical region. Data supporting the changes in cortical region would make the manuscript more acceptable.

7. Osteogenesis and osteoclastogenesis both result in altered serum and urinary marker production. Analysis of differentially regulated markers is suggested. What are the effects on serum levels of bone formation and resorption markers in miR-155 TG and miR-155 KO mice? PINP or Osteoclacin and CTX-1 eliza will be important to study how these mice lose or gain bone.

8. As mentioned by the authors in the abstract that miR155-KO mice show robust bone regeneration in the ectopic and orthotopic models. The authors have not evaluated the pleiotropic effects posed by knocking down mir-155 such as vascular calcification or mineralization. Are there any reports of vital organs being affected in KO mice?

9. Estimation of Expression of RANK and Cathepsin K is also suggested to understand if KO of miRNA- 155 is affecting the osteoclastogenesis process

10. What are the expression levels of miR-155 in bone tissue from miR-155 Tg and miR-155 KO mice? It is important to show miR-155 levels are reduced or knocked down in miR-155 KO bones and increased in miR-155 Tg bones. Authors should mention and explain these effects in the results and discussion.

11. Authors observed no change or possible reduction in Runx2 levels in miR-155 KO BMSCs. However, they observed an increase in Runx2 levels in BMSCs treated with miR-155 sponge. The authors should mention and explain why they did not observe increased Runx2 levels in miR-155 KO BMSCs, in the result and discussion.

12. Authors showed binding site of miR-155-5p in the 3'UTR of S1PR1 gene but did not mention whether it is conserved among different species like mouse, human or rat etc. Authors should comment on these points in the result and discussion.

*Reviewer #1 (Recommendations for the authors):*

1. In the present manuscript the bone microarchitecture at the histological level needs to be analyzed so as to compare the trabecular integrity in both miR155-Tg and miR155-KO mice. The demonstration of S1PR1 expression in a tissue-specific manner in both the groups through immunohistochemistry would validate the hypothesis in an effective way.

2. The study aims at the osteogenic efficacy of miR155, although experiments involving modulation in the resorption levels will support the results convincingly. TRAP staining for the detection of activated osteoclasts at the cellular level will further validate the hypothesis.

3. The manuscript could do with careful editing and typographical correction.

4. Authors are required to provide references for each of the methodologies used in the study.

5. In figure 3, the authors fail to describe the differences observed among the micro CT parameters of the wild-type group in both studies. Please clarify.

6. In figure 4A, authors are advised to represent a good 3D image of calvarial defect where at least the findings can be shown properly.

7. The findings suggest an overall increase in the bone mass in the absence of miR155, but the experiments are restricted to the trabecular region, authors do not mention any variations observed in the cortical region. Data supporting the changes in cortical region would make the manuscript more acceptable.

8. Osteogenesis and osteoclastogenesis both result in altered serum and urinary marker production. A brief analysis of differentially regulated markers is suggested.

9. As mentioned by the authors in the abstract that miR155-KO mice show robust bone regeneration in the ectopic and orthotopic models. The authors have not evaluated the pleiotropic effects posed by knocking down mir-155 such as vascular calcification or mineralization. Are there any reports of vital organs being affected in KO mice?

10. Line 117-119 should be reframed to properly explain the fold increase in micro-CT parameters in KO mice compared to control.

11. Densitometry of western blots should be incorporated by the authors.

12. In figure 7F the loading control i.e., GAPDH is not equal in both the groups. Quantification should be provided.

13. Estimation of Expression of RANK and Cathepsin K is also suggested to understand if KO of miRNA- 155 is affecting the osteoclastogenesis process

*Reviewer #2 (Recommendations for the authors):*

1. Result 6, Page 7, line 195-196, miR-155 was knockdown…….in oetogenic differentiation. Please revise the sentence.

2.Result 7, Page 8, line 208-207, miR-155 robustly enhanced…….in BMSCs. It should be miR-155 sponge instead of miR-155. Please revise the sentence.

3. Authors should revise the figures. Phenotypic results which include microCT data should be in one figure rather than two small different figures. For e.g Figure 1 and 2 should be combined.

*Reviewer #3 (Recommendations for the authors):*

The author could characterize MSCs by flowcytometry post-silencing of miR155, instead of only focusing on how they proliferate and differentiate.

---

## [Author Response]

Essential revisions:In this study entitled " MicroRNA-155 regulates osteogenesis and bone mass phenotype via targeting S1PR1 gene" the authors have shown that miR155 showed a catabolic effect on osteogenesis and bone mass phenotype via interaction with the Sphingosine 1-phosphate receptor-1 (S1PR1) gene. It is also suggested that inhibition of miR155 is a potential strategy for bone regeneration and bone defect healing. It is an important study as cancer/cancer metastasis-induced bone loss-mediated fracture is a serious clinical problem. The authors have shown that miR155 as a potential strategy for bone regeneration and bone defect healing. However, the authors should address the concerns of the reviewers before the manuscript may be accepted for publication.1. As cancer/cancer metastasis-induced bone loss is not gender specific it is important to know if the similar effects are observed in female mice? Authors did experiments in 8 week old male mice. Does this high bone mass phenotype in miR-155 KO or miR-155 Tg mice change over the period of time as mice age? It would be interesting in bone field if these mice are resistant/prone to age related bone loss or inflamaging.

We used a total of 8 mice, of which 4 were male and 4 were female mice. At the 8-week-old age, bone parameters in male and female mice were almost similar, therefore we combined the total 8 mice (male and female) data (Figure 1 and 2). Due to a too-long period required to generate aged mice, we were unable to analyze bone parameters in aged mice. However, we have now performed additional experiments to test the role of *Mir155* on inflammation-related bone loss. These results are now provided in Figure 3. We found that *Mir155*-KO prevents LPS-induced bone loss. The levels of cortical bone and trabecular bone were significantly increased in LPS treated *Mir155*-KO mice compared to wild-type.

2. Bone microarchitecture at the histological level need to be analyzed so as to compare the trabecular integrity in both miR155-Tg and miR155-KO mice. The demonstration of S1PR1 expression in a tissue-specific manner in both the groups through immunohistochemistry would validate the hypothesis in an effective way.

The bone microarchitecture of *Mir155*-Tg and *Mir155*-KO mice at the histological level have now been analyzed and are shown by H&E staining in Figure 1B and Figure 2B, respectively. We performed the immunohistochemistry for S1PR1 using three antibodies from different companies available in China. Unfortunately, these antibodies did not work.

3. The study aims at the osteogenic efficacy of miR155, although experiments involving modulation in the resorption levels will support the results convincingly. TRAP staining for the detection of activated osteoclasts at the cellular level will further validate the hypothesis. Authors should mention and explain these effects in the results and discussion.

We agree that TRAP staining provides an more detailed insight on the effects of *Mir155* on bone resorption. To test the effect of *Mir155* on osteoclast formation, we isolated bone marrow cells from *Mir155*-KO mice and WT mice and induced the cells with M-CSF (50 ng/mL) for 4 days. After that RANKL (50 ng/mL) was added for osteoclastogenic differentiation for 7 days. TRAP staining and osteoclast counting were performed. Information and results have been added to the Materials and methods, results, and Discussion section. The figures of thee experiments are presented in Figure 9. The experiments show that *Mir155*-KO inhibits osteoclastogenesis.

4. In figure 3, the authors fail to describe the differences observed among the micro CT parameters of the wild-type group in both studies. Clarification is required in this regard.

The *Mir155* KO mice were C57BL/6J background, and *Mir155*-Tg mice were FVB background. The differences in Micro CT parameters of the wild-type groups may have resulted from a different background of wild-type mice.

5. In figure 4A, authors are advised to represent a good 3D image of calvarial defect where at least the findings are properly depicted. What are the effects on Dynamic histomorphometry in miR-155 TG and miR-155 KO mice? Calcein/Alizarin Red injections followed by histomorphentry will be important to study Bone formation and mineral apposition rates.

We used the calvarial defect model without adding osteoinductive factors in *Mir155*-KO mice and WT mice. The recovery of bone defect was not very good after 1 month, which may have influenced the 3D image presentation. We have now put these results in Supplementary data. We performed a new experiment, in which BMP2 (180 ng) was applied to the collagen film and grafted into the defect of *Mir155*-KO mice and WT mice. After one month, the samples were collected and analyzed with microCT. The results of microCT and dynamic histomorphometry of this new experiment are shown in Figure 5. Calcein/Alizarin red staining is a good way to detect the dynamic bone formation, however, due to the lack of a hard tissue sectioning instrument in our facility, we were unable to visualize Calcein/Alizarin Red labeling.

6. The findings suggest an overall increase in the bone mass in the absence of miR155, but the experiments are restricted to the trabecular region, authors do not mention any variations observed in the cortical region. Data supporting the changes in cortical region would make the manuscript more acceptable.

We now have also analyzed the cortical region. The results of cortical region analysis are shown in Figure 1C, 1D, 2C, and 2D. These data show that reduced cortical bone level and bone volume/total volume (BV/TV) in *Mir155-Tg* mice. And a higher cortical bone level and BV/TV was shown in *Mir155*-KO mice.

7. Osteogenesis and osteoclastogenesis both result in altered serum and urinary marker production. Analysis of differentially regulated markers is suggested. What are the effects on serum levels of bone formation and resorption markers in miR-155 TG and miR-155 KO mice? PINP or Osteoclacin and CTX-1 eliza will be important to study how these mice lose or gain bone.

As suggested by the reviewer, we collected the serums from *Mir155-*KO mice and WT mice. The PINP and CTX-1 were detected by ELISA. As shown in Figure 9D, the level of PINP was significantly increased while CTX-1 was reduced in *Mir155*-KO mice compared to WT mice, which is consistent with the bone phenotype in vivo. Due to a lack of enough *Mir155*-Tg mice, we were unable to analyze PINP and CTX-1 levels in these mice's serum.

8. As mentioned by the authors in the abstract that miR155-KO mice show robust bone regeneration in the ectopic and orthotopic models. The authors have not evaluated the pleiotropic effects posed by knocking down mir-155 such as vascular calcification or mineralization. Are there any reports of vital organs being affected in KO mice?

Reports from the literature have shown the inhibition of vascular calcification in *Mir155*-KO mice. Therefore, we did not perform this experiment. For better clarification, we added the following text in the Discussion section (page 13, line 366-376): “Clinical application of osteogenic factors in vivo always poses the risk of vascular calcification. miR155-5p overexpression has been reported to aggravate vascular calcification ^(55)^. Importantly, *Mir155*-KO mice also show resistance against vitamin D3-induced vascular calcification ^(56)^. Moreover, *Mir155* deletion inhibits the migration and apoptosis of vascular smooth muscle cells as well as vascular calcification ^(56)^. Furthermore, Zhang et al. showed normal histology of vital organs including the heart, lung, liver, and spleen in *Mir155*-KO mice ^(57)^. These reports from the literature indicate that *Mir155*-KO knockout does not pose the risk of vascular calcification. However, the effect of *Mir155* inhibitors or anti-*Mir155* on vascular calcification should be thoroughly investigated before applying these agents for bone regeneration applications.”

9. Estimation of Expression of RANK and Cathepsin K is also suggested to understand if KO of miRNA- 155 is affecting the osteoclastogenesis process

The expression levels of *Rank* and *Ctsk* were significantly downregulated in *Mir155*-KO BMMs’ osteoclastogenic process. The results are shown in Figure 9C.

10. What are the expression levels of miR-155 in bone tissue from miR-155 Tg and miR-155 KO mice? It is important to show miR-155 levels are reduced or knocked down in miR-155 KO bones and increased in miR-155 Tg bones. Authors should mention and explain these effects in the results and discussion.

The *Mir155* levels were downregulated 0.079 fold in the *Mir155*-KO mice while upregulated 8.57 fold in the miR155-Tg mice bone tissues by RT-qPCR. The results are shown in Figure 1A and Figure 2A. We added the relevant information in the results and Discussion sections.

11. Authors observed no change or possible reduction in Runx2 levels in miR-155 KO BMSCs. However, they observed an increase in Runx2 levels in BMSCs treated with miR-155 sponge. The authors should mention and explain why they did not observe increased Runx2 levels in miR-155 KO BMSCs, in the result and discussion.

We had mentioned this effect in the result section of the original manuscript. We now have added the following text in the Discussion section (page 11, line 295-303): “Whole body knockout of *Mir155* also knock outs the *Mir155* in the other cell types present in bone microenvironment for example monocytes and macrophages. The *Mir155*-KO BMSCs might have certain effects of surrounding *Mir155*-KO other cell types on S1PR1 expression, while the *Mir155* sponged BMSCs from wild-type mice sorted with FACAS may be free from such effects. This could be the possible explanation for the lack of change in RUNX2 expression in *Mir155*-KO BMSCs, but rather an upregulation of RUNX2 expression in *Mir155* sponged BMSCs from wildtype mice observed in this study. However, further studies are required to support this logic.”